# Vitamin D and Its Role in the Lipid Metabolism and the Development of Atherosclerosis

**DOI:** 10.3390/biomedicines9020172

**Published:** 2021-02-09

**Authors:** Andrei Mihai Surdu, Oana Pînzariu, Dana-Mihaela Ciobanu, Alina-Gabriela Negru, Simona-Sorana Căinap, Cecilia Lazea, Daniela Iacob, George Săraci, Dacian Tirinescu, Ileana Monica Borda, Gabriel Cismaru

**Affiliations:** 1Fifth Department of Internal Medicine, Cardiology Clinic, Iuliu Hațieganu University of Medicine and Pharmacy, 400012 Cluj-Napoca, Romania; 2Sixth Department of Medical Specialties, Endocrinology, Iuliu Hațieganu University of Medicine and Pharmacy, 400012 Cluj-Napoca, Romania; oana_pinzariu@yahoo.com; 3Sixth Department of Medical Specialties, Diabetes and Nutritional Diseases, Iuliu Hațieganu University of Medicine and Pharmacy, 400012 Cluj-Napoca, Romania; dana.ciobanu@umfcluj.ro; 4Cardiology Department, “Victor Babes” University of Medicine and Pharmacy, 300041 Timișoara, Romania; eivanica@yahoo.com; 5Pediatric Clinic No 2, Cardiology Department, Iuliu Hațieganu University of Medicine and Pharmacy, 400012 Cluj-Napoca, Romania; cainap.simona@gmail.com; 6Pediatric Clinic No 1, Cardiology Department, Iuliu Hațieganu University of Medicine and Pharmacy, 400012 Cluj-Napoca, Romania; cicilazearo@yahoo.com; 7Pediatric Clinic No 3, Cardiology Department, Iuliu Hațieganu University of Medicine and Pharmacy, 400012 Cluj-Napoca, Romania; iacobdaniela777@gmail.com; 8Internal Medicine Department, Iuliu Hațieganu University of Medicine and Pharmacy, 400012 Cluj-Napoca, Romania; gsaraci@yahoo.com; 9Sixth Department of Medical Specialties, Nephrology, Iuliu Hațieganu University of Medicine and Pharmacy, 400012 Cluj-Napoca, Romania; dacitiri@gmail.com; 10Sixth Department of Medical Specialties, Medical Rehabilitation, Iuliu Hațieganu University of Medicine and Pharmacy, 400012 Cluj-Napoca, Romania; monicampop@yahoo.com; 11Fifth Department of Internal Medicine, Cardiology-Rehabilitation, Iuliu Hațieganu University of Medicine and Pharmacy, 400012 Cluj-Napoca, Romania; gabi_cismaru@yahoo.com

**Keywords:** vitamin D, cardiovascular, atherosclerosis, 25-OH vitamin D, rickets, osteomalacia, bone metabolism, myocardial infarction, ischemic heart disease, ischemic stroke

## Abstract

Vitamin D, a crucial hormone in the homeostasis and metabolism of calcium bone, has lately been found to produce effects on other physiological and pathological processes genomically and non-genomically, including the cardiovascular system. While lower baseline vitamin D levels have been correlated with atherogenic blood lipid profiles, 25(OH)D supplementation influences the levels of serum lipids in that it lowers the levels of total cholesterol, triglycerides, and LDL-cholesterol and increases the levels of HDL-cholesterol, all of which are known risk factors for cardiovascular disease. Vitamin D is also involved in the development of atherosclerosis at the site of the blood vessels. Deficiency of this vitamin has been found to increase adhesion molecules or endothelial activation and, at the same time, supplementation is linked to the lowering presence of adhesion surrogates. Vitamin D can also influence the vascular tone by increasing endothelial nitric oxide production, as seen in supplementation studies. Deficiency can lead, at the same time, to oxidative stress and an increase in inflammation as well as the expression of particular immune cells that play a pivotal role in the development of atherosclerosis in the intima of the blood vessels, i.e., monocytes and macrophages. Vitamin D is also involved in atherogenesis through inhibition of vascular smooth muscle cell proliferation. Furthermore, vitamin D deficiency is consistently associated with cardiovascular events, such as myocardial infarction, STEMI, NSTEMI, unstable angina, ischemic stroke, cardiovascular death, and increased mortality after acute stroke. Conversely, vitamin D supplementation does not seem to produce beneficial effects in cohorts with intermediate baseline vitamin D levels.

## 1. Introduction: Vitamin D Deficiency—A Pandemic; Atherosclerosis—The First Cause of Mortality

Vitamin D, a lipid-soluble cholesterol-based molecule, is synthetized in adequate amounts in people with sufficient sun exposure or, in lesser amounts, taken up through diet. If deficient, the intake can be supplemented with oral formulations [1]. Although presently uncommon in developed countries, vitamin D severe deficiency can cause rickets and osteomalacia in children and adults, respectively. Notwithstanding, less severe subclinical deficiency levels encompassing osteoporosis are more prevalent and associated with the risk of fractures [2]. Although the effects on bone turnover and calcium-phosphate homeostasis are its most widely recognized functions, this molecule acts as a hormone, exerting immunomodulatory actions [3], controlling cellular proliferation and differentiation, and is directly associated with lower risk of obesity, diabetes mellitus, metabolic syndrome, and cardiovascular disease and has neuroprotective and antiaging effects [4,5].

Despite its proven beneficial effects, 25-hydroxyvitamin D [25(OH)D] deficiency is currently considered a European and global pandemic [6,7]. The most acute deficiency strikes low- and middle-income countries (LMICs), where vitamin D deficiency is encountered in 50% to 66% of adults and a staggering 90% to 99% of infants while in the USA, up to 37% of adults and up to 46% of dark-skinned infants suffer from this condition [6]. A 2016 analysis considering mostly Nordic and western Europe populations found significant variability between countries for the percentage of the population classified as deficient: from a low of 6.6% in a Finnish study to a high of 76% in Norwegian studies [7]. This high variability appears to be dependent on the age groups studied. When considering only the adult population, Nordic countries appear to have a lower incidence of 25(OH)D deficiency, most probably due to increased vitamin supplementation or food fortification compared to lower-latitude countries, such as the United Kingdom, Netherlands, and Germany. Consequently, these data are a matter of concern from the point of view of public health.

Atherosclerosis, a chronic condition of the arterial blood vessels, develops as intima internalization of lipids leads to fatty streaks, which in turn evolve to atherosclerotic plaques [8]. The presence of stenotic atherosclerotic lesions leads to chronic tissue ischemia or, if eroded, can produce a thrombus that acutely severs blood flow and produces tissue necrosis. These pathological mechanisms are expressed clinically as ischemic heart disease (IHD), ischemic stroke, or peripheral artery disease (PAD). Thus, atherosclerosis leads to cardiovascular disease (CVD), the first cause of mortality globally. Similar to vitamin D deficiency, atherosclerosis is viewed as a global pandemic, with developed nations having a lower level and low- and middle-income countries (LMICs) having still elevated levels of incidence [9]. Worldwide, almost 30% of individuals with ages between 30 and 79 years, had abnormal carotid intima-media thickness and 21% had carotid atherosclerotic plaques, both a consequence of the atherosclerotic process at the level of the carotid arteries [10].

## 2. Metabolism of Vitamin D

Cholecalciferol, the form of vitamin D named D_3_, is synthetized in the skin from 7-dehydrocholesterol upon irradiation with ultraviolet waves in the range of ultraviolet B light (UV-B) (Figure 1) [11]. 7-dehydrocholesterol is part of the metabolic pathway that controls the synthesis of cholesterol in human cells. By absorbing ultraviolet radiation, which can be ionizing, 7-dehydrocholesterol turns into pre-vitamin D3, which, in turn, because of its molecular instability, converts to cholecalciferol. This process of absorbing UVB takes place in the cellular membrane, and the resulting vitamin D is expelled in the extracellular space, binding to a carrier protein named vitamin D-binding protein. Sunlight also acts as a regulator of vitamin D production in that increased sunlight exposure is captured by pre-vitamin D and vitamin D and transforms these molecules into photoisomers, which do not have any biological activities [12]. Although production of vitamin D_3_ in the skin is the primary source in humans, it can be found in and taken up from food, such as fish oil or mushrooms, in the form of ergocalciferol (Figure 2) [13]. Skin synthesis of vitamin D_3_ rises proportionally with the intensity of the UV radiation. It also reduces proportionally with sunblock usage or the quantity of melanin encountered in the skin, i.e., in higher-latitude-living cohorts, during months with reduced sun exposure, or in patients with darker skin [11,14,15]. However, cholecalciferol is not biologically active; thus, vitamin D is hydroxylated in the liver cells to form 25(OH)D followed by 1α-hydroxylation [11]. The active hormonal form is produced in this last step of 1α-hydroxylation mainly in the kidneys and at other extrarenal sites, resulting in a compound named 1,25(OH)2D_3_ [16,17,18].

## 3. Mechanism of Action

The hormonal form of vitamin D, being a lipid-soluble molecule, is transported in the blood bonded to a serum protein named vitamin D-binding protein (DBP) [19]. At the molecular level, vitamin D in the form of 1,25(OH)2D_3_ exerts its actions by binding to a membrane-bound and cytoplasmic receptor named vitamin D receptor (VDR), which can be found in almost all human tissue, including the cardiovascular system [5,11]. Binding of vitamin D to its VDR is critical for its action because 1,25 dihydroxy vitamin D, the active form, penetrates the cell membrane and binds to VDR [20]. This vitamin D-VDR complex acts with the retinoic acid receptor (RXR) and forms important heterodimers that activate elements of the vitamin D response elements by initiation of the cascade of molecular interactions regulating the suppression and transcription of specific genes [21]. In total, VDR has a direct action on the expression of more than 1000 genes [22]. Overall, vitamin D–VDR interaction decreases the expression of proinflammatory cytokines IL-2 and IL-12 [23]. VDR, being a transcription factor, mediates vitamin D’s genomic actions, controlling a large number of genes, approximately 3% of the genome [13]. At the same time, vitamin D supplementation in insufficient-defined individuals is associated with gene expression. In a 2020 randomized controlled trial, different doses of supplementation, i.e., 600, 2000, and 10,000 IU, and placebo were given to patients with levels of vitamin D of less than 30 ng/mL [24]. After 6 months, there was a dose-dependent increase in serum 25(OH)D and gene expression levels. Conversely, a similar 2018 study did not find any differences in gene expression at 12 months after supplementation with 2000 and 4000 I.U and placebo in patients with baseline levels of approximately 20 ng/mL [25]. Ways in which Vitamin D acts non-genomically have also been identified, such as through intracellular signaling molecules, generation of second messengers, and activation of specific protein kinases [26]. The change in the chemical structure of cholecalciferol led to the emergence of new molecules, which, surprisingly, have the ability to bind to VDR. A number of products are presented in Table 1 together with their properties.

## 4. Normal Values

It is recommended to evaluate vitamin D levels by using serum 25(OH)D levels expressed in ng/mL or nmol/L. One unit of ng/mL is roughly equivalent to 2.5 units of nmol/L [27]. Multiple threshold values for vitamin D deficiency in non-pregnant adults have been proposed throughout the decades [28]. Most authorities define deficiency at levels lower than 20 ng/mL and insufficiency at levels between 21 and 29 ng/mL while recommended values are considered to be higher than 30 ng/mL or 75 nmol/L [27,28,29]. Although it is suggested that values more than 30 ng/mL would be sufficient, there is no clear consensus on the optimal values of serum vitamin D. Interrelation studies of 25(OH)D with muscle strength, and risk of breast cancer or colon cancer have urged several investigators to suggest an optimal 25(OH)D target for multiple health outcomes: 40 to 50 ng/mL [30,31]. Garland et al., based on the epidemiology of cancer, suggest that increasing 25(OH)D to a 40 to 60 ng/mL range could reduce breast cancer risk by 25% and colon cancer risk by 27% [32].

## 5. Vitamin D’s Action on Lipid Profile

Deficient vitamin D levels corelated with unfavorable serum lipid levels and sufficient values with good lipid profiles in observational and interventional studies [33]. More recent studies confirm these findings [34,35,36]. In a 2016 study examining a Polish cohort of patients, there was an inverse relationship between vitamin D levels and total cholesterol (TC), low-density-lipoprotein cholesterol (LDL-C), and triglycerides (TGs) [34]. A study analyzing the levels of 25(OH)D and various lipid fractions of more than 20,000 showed a statistically significant correlation between vitamin D deficiency and an atherogenic lipid profile of the patients [35]. Moreover, recently, meta-analyses have been published evaluating vitamin D levels and vitamin D supplementation and their correlation with the lipid profile [37,38,39]. In one 2015 meta-analysis of eight randomized controlled trials (RCTs) evaluating the impact of vitamin D supplementation on the lipid profile found a lowering effect for TG and a correlation with increasing high-density-lipoprotein cholesterol (HDL-C) but also with LDL-C [37]. The interpretation of these results should be cautionary because of the reduced number of studies considered and the high heterogeneity of outcomes and interventions, i.e., the dose of vitamin D supplementation. In a much larger meta-analysis evaluating the pooled effect of vitamin D supplementation on TG, TC, LDL-C, and HDL-C in up to 39 RCTs, there was an inverse and statistically significant relationship between vitamin D supplementation and TG, TC, and LDL-C. In contrast, vitamin D supplementation increased levels of HDL-C [38]. Similarly, in a meta-analysis of seven RCTs of co-supplementation of vitamin D and calcium in overweight and obese patients, the authors found that less than 8 weeks of supplementation led to a statistically significant TG, TC, and LDL-C reduction and an increase in HDL-C levels [39]. Low levels of vitamin D have been identified in polycystic ovary syndrome (PCOS), especially in obese patients and those with a waist-to-hip ratio above 0.85 [40]. Additionally, vitamin D deficiency may be a risk factor in the development of this endocrine-metabolic pathology [41]. A meta-analysis of 11 RCTs that included 483 patients with PCOS evaluated the effect of vitamin D supplementation versus placebo [42]. The group treated with vitamin D supplements showed a reduction in insulin resistance and TC versus the placebo group. However, vitamin D supplements did not improve HDL-C and TG levels in patients with PCOS.

In children, supplementation of vitamin D to maintain serum 25(OH)D in the optimal range is associated with a lower risk for developing type 1 diabetes and also helps to control the disease activity [43].

## 6. Vitamin D’s Action on Atherosclerosis

Cardiovascular disease (CVD) is the main cause of mortality globally and atherosclerosis is the prime step in CVD development. While historically atherosclerosis was thought to actualize through an accumulation of lipids in the blood vessel walls, the current position is that this condition involves the immune system cells and the process of inflammation and circulating cholesterol molecules [44]. The atherosclerosis process begins as an interplay between blood vessel wall injury, inflammation, endothelial dysfunction, and immune cell recruitment [45]. The risk factors for atherosclerosis include hyperlipidemia, cigarette smoking, hypertension, diabetes, and ageing, leading to endothelial cell (EC) dysfunction and recruitment and stimulation of inflammatory and smooth muscle cell (SMC) proliferation. Typically, after these first steps, lipoproteins start accumulating in the vessel wall and platelets and monocytes adhere to the intima layer. The internalized monocytes transform into macrophages and later into foam cells. SMCs are recruited and then proliferate in the extracellular matrix (ECM) produced along with T cell recruitment. Then, lipids are accumulated intracellularly and in the ECM, forming the characteristic atherosclerotic plaque. New data about atherosclerosis’s pathophysiology has become available during the last decades showing that vitamin D plays a particular role in developing this condition [46,47]. Early atherosclerosis in children was evidenced by the Bogalusa Heart Study and a large number of cardiovascular risk factors were found to associate with atherosclerosis in children. The progression of the atherosclerotic process in early life is associated with obesity, alteration in the lipid profile, and increased blood pressure values. There is evidence that low vitamin D levels in children are associated with increased blood pressure and dyslipidemia and the anti-inflammatory effect of vitamin D could ameliorate these cardiovascular risk factors [48].

### 6.1. Endothelial Adhesion and Activation

Initial murine and human randomized trials showed the possible link between vitamin D insufficiency or vitamin D inaction, i.e., in VDR-null mice and the increased vascular cell adhesion levels in humans or a state of endothelial cell activation in vivo in mice [49,50]. Similarly, in a 2012 study, it was found that in patients with type 2 diabetes mellitus (T2DM), vitamin D deficiency was associated with increased expression of the M2 monocyte phenotype, which expresses surface molecules aiding in adhesion to endothelial cells [51]. In a 2017 RCT, Salekzamani et al. proved that in patients with metabolic syndrome, a weekly dose of vitamin D of 50,000 IU for 16 weeks significantly increased the level of vitamin D from insufficient to sufficient and significantly reduced the levels of cytokine IL-6, vascular cell adhesion molecule 1 (VCAM-1), and E-selectin, which are known to be involved in the adhesion process of the endothelial cells [52]. Similarly, Szeto et al. provide evidence that VDR-null mice exhibit increased serum levels of ICAM-1 and MCP-1, compounds known as adhesion molecules [53]. Moreover, the macrophages lacking VDR present an increase in the uptake of cholesterol and this results in cholesterol accumulating in the intracellular space. This study also showed that VDR-null macrophages exhibit increased renin-angiotensin system components, such as renin, angiotensinogen, or AT1R, and these animal subjects had significantly higher levels of atherosclerosis. Blocking renin with aliskiren ameliorated atherosclerosis. Thus, Szeto et al. showed that the antiatherogenic effects of vitamin D allegedly lie with the inhibition of RAS in the immune cells that are involved in the process of atherosclerosis.

### 6.2. Vascular Tone and Endothelial Function

The vascular tone is most commonly controlled through nitric oxide (NO). NO is produced in humans through NO synthase (NOS), which has several different isoforms, and the endothelial form (eNOS) is the one present constitutively in the blood endothelial cells [54]. Through its actions on the vessels’ endothelial cells, vitamin D increases the nitric oxide levels, which is known to be a vasodilator and platelet antiaggregant and reduces the level of reactive oxygen species released [55,56]. eNOS activity through its control of vascular constriction or dilatation is one of several ways that endothelial function manifests. Altered endothelial function results in vascular tone abnormalities, increased permeation of molecules into the endothelial cells, and increases in pro-coagulation and/or prothrombotic activity [46]. While vitamin D deficiency correlated in some particular instances with endothelial dysfunction [57], recent meta-analyses of RCTs did not find a definite connection between vitamin D supplementation and endothelial function (EF) [58,59]. A 2017 meta-analysis of approximately 1200 patients evaluating the outcome of vitamin D supplementation on endothelial function did not find any statistically significant correlation between patient groups. Endothelial function was evaluated mainly through flow-mediated dilatation (FMD), pulse-wave velocity (PWV), or augmentation index (AI), which are recognized surrogates of EF [58,60,61,62]. Similarly, in one 2018 meta-analysis evaluating endothelial function indirectly through FMD, PWV, or AI in response to vitamin D supplementation in approximately 1800 patients with metabolic syndrome or related conditions, a statistically significant correlation only in FMD improvement without affecting PWV or AI was found [60]. Additional data comes from a small population RCT from 2020, where 4 weeks of 2000 IU/day vitamin D supplementation in an African American cohort significantly reduced oxidative stress and increased endothelial nitric oxide production and bioavailability [63]. These two mechanisms are known to give rise to atherosclerotic disease [57]. In this trial, African American participants had median serum vitamin D levels below that of the recommended levels of 50 nmol/L established by European or International authorities presented previously [28,29]. It must be specified that these results were not observed in the comparison cohort of European Americans, which in turn had higher baseline levels of serum vitamin D so that these beneficial effects of vitamin D supplementation could be observed only in deficient patients.

### 6.3. Inflammation and Atherosclerosis

The pathological process of atherosclerosis is presently known to develop through chronic inflammation of the blood vessel wall brought about by the innate immune response cells, such as monocytes or macrophages [64]. As described previously, the process of atherosclerosis involves the innate immune system in the form of monocytes and macrophages that invade the vascular intima and a process of inflammation. Although most of the cells involved in the atherosclerotic plaque are macrophages, a limited role is also played by B and T cells, which are considered part of the adaptive immune system [65]. Modulating the immune and inflammatory response in atherogenesis could provide potential future clinical benefits besides the classical lipid-lowering approach [66]. Vitamin D has been found to act as an anti-inflammatory mediator in certain instances, which incurs the possibility of its benefit in lowering atherosclerosis development [67]. Vitamin D appears to reduce proinflammatory type 1 cytokines: IL-12, IL-6, IL-8, IFN-gamma, and TNF-alpha; and increase anti-inflammatory type 2 cytokines: IL-4, IL-5, and IL-10 (Figure 3) [68,69]. In patients with rheumatoid arthritis, a condition of increased inflammation associated with endothelial dysfunction, premature atherosclerosis, and a higher risk of developing CVD, it has been found that vitamin D levels are low and corelate with the disease [70]. In children, where vitamin D deficiency and insufficiency is much more prevalent than in adults, there has been a link between vitamin D deficiency and oxidative stress and inflammation through surrogate molecules, such as cathepsin, IL-6, and adiponectin [71]. Furthermore, an association was reported in the pediatric population between vitamin D deficiency and incidence and severity of viral respiratory tract infection. Vitamin D antiviral activity relies on the suppression of the overexpression of proinflammatory cytokines and modulation of natural killer cells’ function and toll-like receptor expression [43].

Based on their previous data on atherosclerosis development in murine models, Amy et al. proved in a 2017 RCT that vitamin D supplementation in patients diagnosed with type 2 diabetes mellitus (T2DM) decreases the levels of LDL cholesterol uptake by circulating monocytes, one of the first steps that leads to the development of atherosclerosis [72]. Previously, it was shown that in mice, vitamin D-induced deficiency in monocytes and macrophages leads to the increase of the transport of cholesterol into the blood vessel wall by monocytes and to cholesterol uptake by atherosclerotic plaque macrophages, respectively [73]. Similarly, it appears that in patients with T2DM, vitamin D deficiency was associated with increased expression of the M2-monocyte phenotype, which expresses surface molecules facilitating adhesion to endothelial cells [52]. Building on these data, the authors proved that in patients with T2DM without disease-specific complications and with median baseline levels of vitamin D of less than 25 ng/mL, supplementation with 4000 I.U. of vitamin D daily for 4 months reduced total cholesterol contained within monocytes. However, intervention did not lead to lower levels of triglycerides, TC, LDL, or HDL cholesterol [72]. These findings suggest that vitamin D supplementation could inhibit atherosclerosis progression. In the context of these findings, there is a need for randomized controlled trials documenting this process in the general population. Despite this early data, vascular inflammation has not yet been decisively associated with vitamin D deficiency or vitamin D supplementation [74,75].

### 6.4. Arterial Smooth Muscle Cells

Since the smooth muscle cells play an important role in the development of atherosclerosis through proliferation inside the pathological fibrous and fatty atheroma [42], amelioration of this pathological process could have an impact on atherogenesis. The vitamin D receptor is known to be found on the endothelial cells’ surface and vitamin D produces effects by binding to its receptor [76]. Additionally, VDR is encountered in arterial SMC based on data in rabbits that prove calcium uptake into the muscle cells coupled with degeneration of SMC in moments of very high serum vitamin D levels [77]. In rat models of cultured aortic SMC, it has been shown that the lack of vitamin D signaling in VDR knock-out (VDRKO) mice led to increased production of angiotensin II, which, in turn, is known to lead to increased oxidative stress and premature cellular ageing known as cellular senescence [78]. This data could explain the previously proved association of vitamin D deficiency and increased cardiovascular risk as blocking angiotensin II’s actions is for a long time one of the cornerstones in many cardiovascular therapeutics [79,80]. In a similar rat model of SMC, it was shown that 1,25(OH)D determines the decrease of smooth muscle proliferative activity non-genomically by inhibiting endothelin (ET) and an ET-activated cyclin-dependent kinase 2, one key kinase with an essential role in the cell cycle [81]. Another study of aortic SMC in a mouse model reported that vitamin D modulates tissue factor (TF) expression, genomically reducing its transcription through the activity of nuclear factor kappa-light-chain-enhancer of activated B cells (NF-κB) [82]. TF is a known agent involved in the process of clotting and if vitamin D has similar functions in human SMC, vitamin D sufficiency or supplementation could provide a benefit and possibly explain at least one mechanism through which vitamin D is associated with lower CV risk [83].

All the actions of vitamin D on atherosclerosis presented previously are summarized in Table 2.

## 7. Clinical Studies on Atherosclerosis and Vitamin D Deficiency and Vitamin D Supplementation

The role of vitamin D deficiency or supplementation in the development of myocardial infarction or stroke has been evaluated in recent years and has brought into focus new important data [84,85,86]. In an observational study of approximately 340 patients, those with lower levels of vitamin D were found to be slightly more likely to suffer from coronary atherosclerosis as evaluated by coronary computed tomography angiography while the median of these patients’ levels was below those defined as deficient, respectively 20 ng/mL [87]. In an observational study evaluating vitamin D deficiency in 637 patients that were evaluated through coronary catheterization, there was a correlation between increasing vitamin D levels and a lowered number of significant coronary artery lesions [34]. Another small observational retrospective study evaluated vitamin D levels in patients presenting to the emergency room with acute coronary syndromes (ACS), and found that lower levels of vitamin D corelated positively with the presence of ACS and inversely with high-sensitive cardiac troponin T levels [88]. One prospective study comparing patients with ST elevation myocardial infarction (STEMI), non-STEMI (NSTEMI), or unstable angina with a control group without ACS found a statistically significant link between low levels of vitamin D in the study group and the presence of ACS. In contrast, the levels in the control group were significantly higher [89]. In this study, the percentage of patients suffering from an episode ACS with insufficient levels of vitamin D was a staggering 72% while 89% of the same total of patients had deficient serum levels of vitamin D, i.e., lower than 30 ng/mL. In comparison, the control group was 27.5% insufficient and 41.6% deficient in vitamin D. In yet another study of Polish patients diagnosed with uncomplicated acute myocardial infarction (AMI), the group suffering from AMI (*n* = 59) was entirely vitamin D insufficient, with every participant having levels lower than 30 ng/mL and almost 90% of this group being deficient in vitamin D with levels lower than 20 ng/mL [90]. Data that links vitamin D deficiency and ischemic stroke are also available. Analyzing the recorded data from over 10,000 evaluated patients from the prospective Copenhagen City Heart Study, Brøndum-Jacobsen et al. found decreasing lower levels of 25(OH)D were correlated with an increased risk of suffering from an ischemic stroke [91]. This study showed that patients in the 50th to 100th percentile of vitamin D levels had a median of 62 nmol/L while the patients in the lower 25th percentile had a level of 19 nmol/L. Comparing both categories of patients, it came out that those in the 25th percentile had a statistically significant hazard ratio for ischemic stroke of 1.45 (1.16–1.80; 95% confidence interval), an increase of 45%. Another prospective study evaluating and comparing the levels of vitamin D in patients who presented to the emergency room with an ischemic stroke to a control group found that in the stroke group (*n* = 168), vitamin D median levels were significantly lower. Comparing the two groups, 43% of stroke victims were deficient, while only 6% of the control group had vitamin D levels below 20 ng/mL [92]. Another study from Poland evaluated vitamin D levels and mortality in 240 patients after diagnosing ischemic stroke. This study showed that in patients with ischemic stroke, only 1.3% had optimal levels of vitamin D of higher than 30 ng/mL and during the 45-month follow-up, the patients with severe deficiency, defined as levels lower than 10 ng/mL, had a much higher and statistically significant mortality rate per year: 4.81% compared with 1.89% in patients without severe deficiency [93]. Thus, severe vitamin D deficiency is a significant factor that increases death risk in patients with ischemic stroke.

Studies on vitamin D supplementation to reduce CVD burden did not provide positive results. In a 2020 RCT, among randomized patients with mean age of 75, vitamin D supplementation at 2000 IU/day for 3 years did not reduce the initial elevated systolic or the diastolic blood pressure, which in almost all cases was increased due to atherosclerotic disease [94]. Patients in this study had a median vitamin D level of approximately 22 ng/mL and approximately 40% of the patients with supplementation were below the threshold level for insufficiency. This result would suggest that vitamin D supplementation for less than 3 years in the older general population does not influence SBP and DBP. Elevated arterial blood pressure (BP) is a known cause for atherosclerosis; in many animal models, it has been linked to loss of carotid arterial distensibility and has been reviewed as a risk factor for atherosclerosis. Simultaneously, one of the risk factors for developing atherosclerosis is high BP, forming a possible positive feedback loop between these two CV conditions [95,96,97,98]. Similarly, a 2017 RCT evaluating monthly high-dose (100,000 IU) vitamin D supplementation did not find any benefit on cardiovascular disease, including hypertension or angina, direct consequences of systemic and coronary atherosclerosis [99]. The patients evaluated were mainly of European race with a median age of 66 years old, were followed for a median period of 3.3 years, and had a median vitamin D level of 25 ng/mL, although approximately 30% of them were below the insufficiency threshold of 20 ng/mL. The RECORD trial (Randomized Evaluation of Calcium or Vitamin D) showed no effect of low-dose vitamin D supplementation (400–800 IU) on 2–7-year cardiovascular mortality and all-cause mortality [100]. Furthermore, the Women’s Health Initiative showed a non-significant harmful effect of vitamin D supplementation and calcium on non-fatal myocardial infarction, death by coronary artery disease, and need of vascularization [101]. Finally, in another extensive study of almost 26,000 patients who underwent randomization, a daily dose of 2000 IU of vitamin D coupled with daily 1 g of omega-3 fatty acids did not reduce cardiovascular events defined as myocardial infarction, stroke, or death from cardiovascular disease in a median follow-up of 5.3 years [102]. In this study, only 12.7% of patients had 25(OH)D levels below those for insufficiency of 20 ng/mL. All these data do not support the role of vitamin D supplementation in reducing cardiovascular burden of disease in these patients with relatively normal baseline levels of 25(OH)D. While attaining normal levels of vitamin D is recommended in patients that are deficient or insufficient, additional RCTs may be necessary to conclusively evaluate the role of vitamin D supplementation in reducing myocardial infarction, stroke, or overall CVD burden in deficient and/or insufficient patients.

While vitamin D deficiency and insufficiency could represent an important cardiovascular risk factor, there is a need for additional extensive studies analyzing the clinically beneficial role of vitamin D supplementation in patients with low baseline blood levels.

## 8. Future Directions

To date, vitamin D, a highly important hormone in the homeostasis and metabolism of calcium bone, has lately been found to produce effects on other physiological and pathological processes, including the cardiovascular system.

While lower baseline vitamin D levels have been found to corelate with atherogenic blood lipid profiles, 25(OH)D supplementation has been found to influence the levels of serum lipids in that it lowers the levels of total cholesterol, triglycerides, and LDL cholesterol and increases the levels of HDL cholesterol.

To detect whether a true association between vitamin D and atherosclerosis exists, future interventional studies should be designed with sufficient sample size, long follow-up duration, higher vitamin D doses, restriction of lipid-lowering drug use, and potentially control for confounders of atherosclerosis (such as metabolic and endocrine diseases).

Vitamin D is also involved in the development of atherosclerosis at the site of the blood vessels. Deficiency of this vitamin has been found to lead to an increase of adhesion molecules or endothelial activation and at the same time supplementation is linked to the lowering presence of adhesion surrogates. Vitamin D also influences the vascular tone by increasing endothelial nitric oxide production as seen in supplementation studies. A better understanding of these queries might have significant implications for the potential use of vitamin D analogues in the prevention of atherosclerosis.

Vitamin D deficiency is consistently associated with cardiovascular events, such as myocardial infarction, STEMI, NSTEMI, unstable angina, ischemic stroke, cardiovascular death, and increased mortality after the acute faze of stroke. Conversely, vitamin D supplementation does not seem to produce beneficial effects in cohorts with all intermediate baseline levels of vitamin D.

Future work is needed to ascertain whether vitamin D therapy is effective in reducing mortality through reducing the incidence of acute coronary syndromes. Vitamin D might be deficient in patients who smoke, have high blood lipids, diabetes, or other specific risk factors.

## 9. Conclusions

Vitamin D has lately been found to modify the mechanisms of atherosclerosis. Supplementation in deficient patients has been found to lower the levels of total cholesterol, triglycerides, and LDL cholesterol and increase the levels of HDL cholesterol. Vitamin D deficiency increases vascular cell adhesion molecules and E-selectin, with a potential role in the formation of atheroma plaque. It also influences the vascular tone by modifying endothelial nitric oxide production. Deficiency can lead to oxidative stress, increased inflammation, and expression of immune cells, i.e., monocytes and macrophages, that play a pivotal role in atherosclerosis of the intima. Another pathway by which vitamin D is involved in atherogenesis is through inhibition of vascular smooth muscle cell proliferation. Furthermore, studies have shown that vitamin D deficiency is associated with cardiovascular events, such as myocardial infarction, STEMI, NSTEMI, unstable angina, ischemic stroke, cardiovascular death, and increased mortality, after the acute stroke.

## Figures and Tables

**Figure 1 biomedicines-09-00172-f001:**
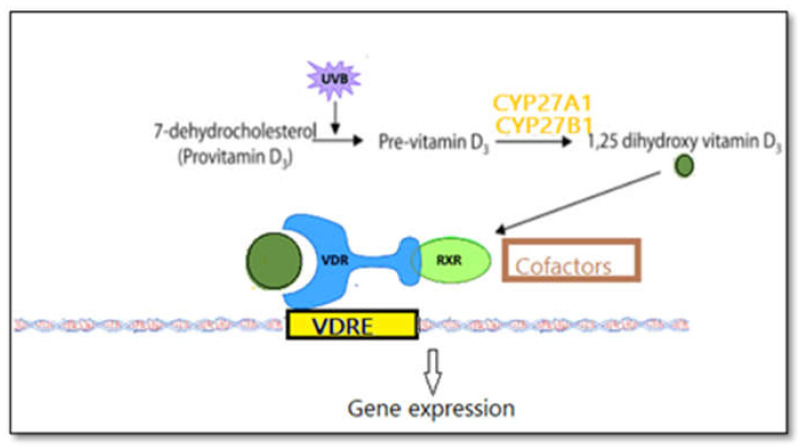
Vitamin D synthesis in the skin.

**Figure 2 biomedicines-09-00172-f002:**
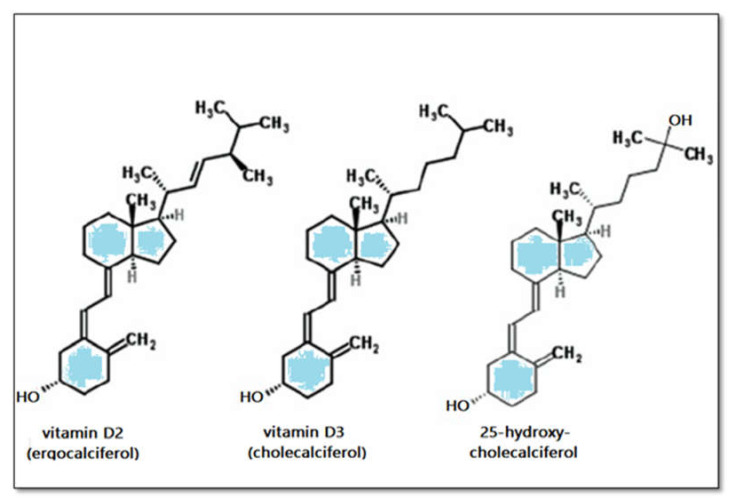
Structure of vitamin D and its derivates.

**Figure 3 biomedicines-09-00172-f003:**
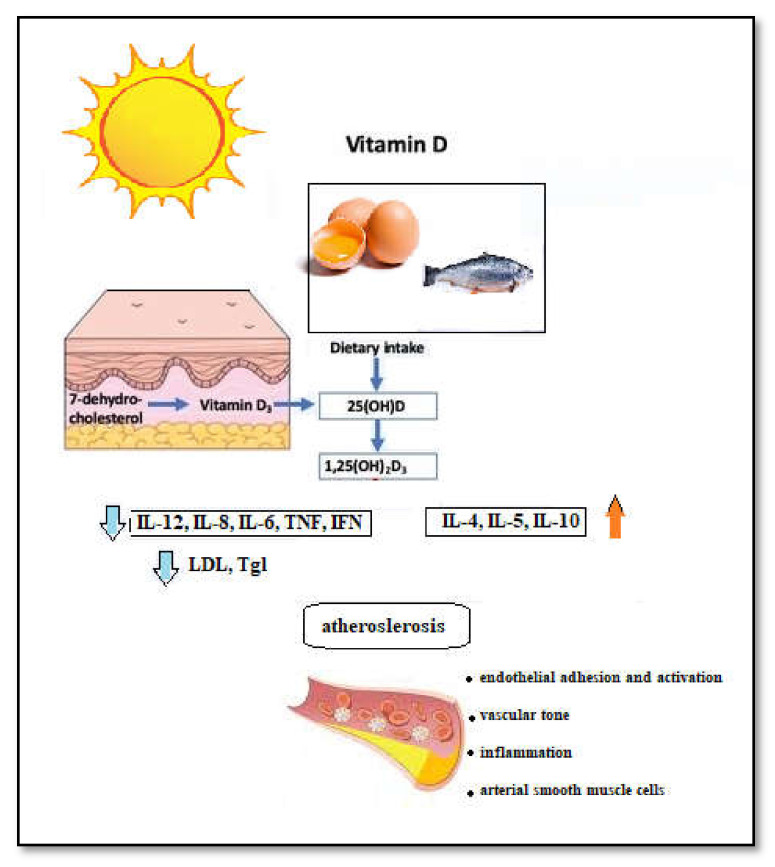
Mechanism of action of vitamin D on the process of atherosclerosis [68,69].

**Table 1 biomedicines-09-00172-t001:** Derivates of vitamin D with clinical implications.

Calcidiol	3,25 (OH)D_3_	Treatment of Renal Osteodystrophy
Calcitriol	4,1,25 (OH)	Treatment of renal osteodystrophy
Calcipotriol	5, 22-ene-26,27-dehydro-1,25(OH)2D_3_	Treatment of psoriasis
Doxercalciferol	6, 1α(OH)D_2_	Treatment of secondary hyperparathyroidism
Alfacalcidol	7, 1α(OH)D_3_	Treatment of osteoporosis
Tacalcitol	8, 1α,24(OH)2D_3_	Treatment of psoriasis
Oxacalcitriol	10, 22-oxa-1,25(OH)2D_3_	Treatment of psoriasis
Falecalcitriol	11, 1,25(OH)2-26,27-F6-D_3_	Treatment of secondary hyperparathyroidism

**Table 2 biomedicines-09-00172-t002:** Vitamin D effects on atherosclerosis.

Lipid profile	reduces total cholesterol reduces LDL-C reduces triglycerides increases HDL-C
Endothelial adhesion and activation	○ reduces vascular cell adhesion molecule 1 (VCAM-1) ○ reduces E-selectin
Vascular tone and endothelial function	increases the level of nitric oxide reduces the level of reactive oxygen species released
Inflammation and atherosclerosis	○ reduces proinflammatory type 1 cytokines: IL-12, IL-6, IL-8, IFN-gamma, TNF-alpha ○ increase anti-inflammatory type 2 cytokines: IL-4, IL-5, and IL-10 ○ reduces oxidative stress through reducing cathepsin, IL-6 and adiponectin
Arterial smooth muscle cells	decreases production of angiotensin II decreases oxidative stress inhibits cellular senescence reduces tissue factor expression

## Data Availability

Not applicable.

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
