# Peer review of "Vitamin D and Its Role in the Lipid Metabolism and the Development of Atherosclerosis"

_biomedicines, 2021, doi:10.3390/biomedicines9020172_

Round 1

Reviewer 1 Report

This review gives a comprehensive summary of recent studies on the role of Vitamin D on lipid metabolism in normal and disease state. The article is well written with clarity on different topics. One minor suggestion for the authors:

The main conclusion is that there are no significant beneficiary effect of Vitamin D supplements on the prevention of atherosclerosis. However, endogenous vitamin D clearly show beneficial effects on lipid metabolism and prevention of events associated with cardiovascular disease. Since the supplements did not show the desired effect, it would be useful to add a small section on future direction of this work.

Reviewer 2 Report

In this manuscript, Surdu et al. reviewed the current scientific literature about vitamin D's role in atherosclerosis and its risk factors. The authors described its metabolism, mode of action, and vitamin D effect on lipid profile and atherosclerosis. In the last part of the review, the authors provide data from clinical studies, supporting the hypothesis that vitamin D plays a role in atherosclerosis development. Overall, the manuscript is well written, up to date, and only minor revision is needed.

  1. In the paragraph "Metabolism of vitamin D" (page 2), the authors are asked to add a figure with the structure of vitamin D and its derivatives and elaborate on the regulation of vitamin D synthesis by UV at the molecular level.
  2. In the section "Mechanism of action" (page 3):Line 112: Is the VDR:RXR complex more active when the RXR ligand binds to the heterodimer?Lines 119-124: what could be the reasons for the discrepancy between the two trials?
  3. Line 120: is RCT = randomized clinical trial?
  4. Line 110: please change 1.25 to 1,25
  1. The section on vitamin's action on atherosclerosis is obligatory to review the experiments performed in animal models. (see Mol Endocrinol. 2012 Jul;26(7):1091-101 and others)
  2. I would recommend adding a section about synthetic vitamin D derivatives.
  3. I would recommend adding a table summarizing the data presented in the review.
